# Dynamics of gaze control during prey capture in freely moving mice

Angie M Michaiel, Elliott TT Abe, Cristopher M Niell*

Institute of Neuroscience and Department of Biology, University of Oregon, Eugene, United States

**Abstract** Many studies of visual processing are conducted in constrained conditions such as head- and gaze-fixation, and therefore less is known about how animals actively acquire visual information in natural contexts. To determine how mice target their gaze during natural behavior, we measured head and bilateral eye movements in mice performing prey capture, an ethological behavior that engages vision. We found that the majority of eye movements are compensatory for head movements, thereby serving to stabilize the visual scene. During movement, however, periods of stabilization are interspersed with non-compensatory saccades that abruptly shift gaze position. Notably, these saccades do not preferentially target the prey location. Rather, orienting movements are driven by the head, with the eyes following in coordination to sequentially stabilize and recenter the gaze. These findings relate eye movements in the mouse to other species, and provide a foundation for studying active vision during ethological behaviors in the mouse.

## Introduction

Across animal species, eye movements are used to acquire information about the world and vary based on the particular goal (*Yarbus, 1967*). Mice, a common model system to study visual processing due to their genetic accessibility, depend on visual cues to successfully achieve goal-directed tasks in both artificial and ethological freely-moving behavioral paradigms, such as the Morris water maze, nest building, and prey capture; (*Morris, 1981*; *Clark et al., 2006*; *Hoy et al., 2016*). It is unclear, however, how mice regulate their gaze to accomplish these visually mediated goals. Previous studies in both freely moving rats and mice have shown that eye movements largely serve to compensate for head movements (*Wallace et al., 2013*; *Payne and Raymond, 2017*; *Meyer et al., 2018*; *Meyer et al., 2020*), consistent with the vestibulo-ocular reflex (VOR) present in nearly all species (*Straka et al., 2016*). While such compensation can serve to stabilize the visual scene during movement, it is not clear how this stabilization is integrated with the potential need to shift the gaze for behavioral goals, particularly because mice lack a specialized central fovea in the retina, and also have laterally facing eyes resulting in a relatively limited binocular field (roughly 40° as opposed to 135° in humans [*Dräger, 1978*]). In addition, because eye movements are altered in head-fixed configurations due to the lack of head movement (*Payne and Raymond, 2017*; *Meyer et al., 2020*), understanding the mechanisms of gaze control and active visual search benefits from studies in freely moving behaviors.

Prey capture can serve as a useful paradigm for investigating visually guided behavior. Recent studies have shown that mice use vision to accurately orient towards and pursue cricket prey (*Hoy et al., 2016*), and have begun to uncover neural circuit mechanisms that mediate both the associated sensory processing and motor output (*Hoy et al., 2019*; *Shang et al., 2019*; *Zhao et al., 2019*; *Han et al., 2017*). Importantly, prey capture also provides a context to investigate how mice actively acquire visual information, as it entails identifying and tracking a localized and ethological sensory input during freely moving behavior. Here, we asked whether mice utilize specific eye movement strategies, such as regulating their gaze to maximize binocular overlap, or actively targeting

*For correspondence:
cniell@uoregon.edu

**Competing interests:** The authors declare that no competing interests exist.

**eLife digest** As you read this sentence, your eyes will move automatically from one word to the next, while your head remains still. Moving your eyes enables you to view each word using your central – as opposed to peripheral – vision. Central vision allows you to see objects in fine detail. It relies on a specialized area of the retina called the fovea. When you move your eyes across a page, you keep the images of the words you are currently reading on the fovea. This provides the detailed vision required for reading.

The same process works for tracking moving objects. When watching a bird fly across the sky, you can track its progress by moving your eyes to keep the bird in the center of your visual field, over the fovea. But the majority of mammals do not have a fovea, and yet are still able to track moving targets. Think of a lion hunting a gazelle, for instance, or a cat stalking a mouse. Even mice themselves can track and capture insect prey such as crickets, despite not having a fovea. And yet, exactly how they do this is unknown. This is particularly surprising given that mice have long been used to study the neural basis of vision.

By fitting mice with miniature head-mounted cameras, Michaiel et al. now reveal how the rodents track and capture moving crickets. It turns out that unlike animals with a fovea, mice do not use eye movements to track moving objects. Instead, when a mouse wants to look at something new, it moves its head to point at the target. The eyes then follow and 'land' on the target. In essence, head movements lead the way and the eyes catch up afterwards.

These findings are consistent with the idea that mammals with large heads evolved eye movements to overcome the energy costs of turning the head whenever they want to look at something new. For small animals, moving the head is less energetically expensive. As a result, being able to move the eyes independent of the head is unnecessary. Future work could use a combination of behavioral experiments and brain recordings to reveal how visual areas of the brain process what an animal is seeing in real time.

and tracking prey. Alternatively, or in addition, mice may use directed head movements to target prey, with eye movements primarily serving a compensatory role to stabilize the visual scene.

Predators typically have front-facing eyes which create a wide binocular field through the overlap of the two monocular fields, allowing for depth perception and accurate estimation of prey location (*Cartmill, 1974*). Prey species, in contrast, typically have laterally facing eyes, and as a result, have large monocular fields spanning the periphery, which allow for reliable detection of approaching predators. Though mice possess these characteristics of prey animals, they also act as predators in pursuing cricket prey (*Hoy et al., 2016*). How then do animals with laterally placed eyes target prey directly in front of them, especially when these targets can rapidly move in and out of the narrow binocular field? This could require the modulation of the amount of binocular overlap, through directed lateral eye movements, to generate a wider binocular field, such as in the case of cuttlefish (*Feord et al., 2020*), fish (*Bianco et al., 2011*), many birds (*Martin, 2009*), and chameleons (*Katz et al., 2015*). In fact, these animals specifically rotate their eyes nasally before striking prey, thereby creating a larger binocular zone. However, it is unknown whether mice use a similar strategy during prey capture. Alternatively, they may use coordinated head and eye movements to stabilize a fixed size binocular field over the visual target.

Foveate species make eye movements that center objects of interest over the retinal fovea, in order to use high acuity vision for complex visual search functions including identifying and analyzing behaviorally relevant stimuli (*Hayhoe and Ballard, 2005*). Importantly, afoveate animals (those lacking a fovea) represent a majority of vertebrate species, with only some species of birds, reptiles, and fish possessing distinct foveae (*Harkness and Bennet-Clark, 1978*), and among mammals, only simian primates possessing foveae (*Walls, 1942*). It remains unclear whether mice, an afoveate mammalian species, actively control their gaze to target and track moving visual targets using directed eye movements, or whether object localization is driven by head movements. We therefore aimed to determine the oculomotor strategies that allow for effective targeting of a discrete object, cricket prey, within the context of a natural behavior.

Recent studies have demonstrated the use of head-mounted cameras to measure eye movements in freely moving rodents (*Wallace et al., 2013*; *Meyer et al., 2018*; *Meyer et al., 2020*). Here, we used miniature cameras and an inertial measurement unit (IMU) to record head and bilateral eye movements while unrestrained mice performed a visually guided and goal-directed task, approach and capture of live insect prey. We compared the coordination of eye and head movements, as well as measurements of gaze relative to the cricket prey during approach and non-approach epochs, to determine the oculomotor strategies that mice use when localizing moving prey.

## Results

### Tracking eye and head movements during prey capture

Food-restricted mice were habituated to hunt crickets in an experimental arena, following the paradigm of *Hoy et al., 2016*. To measure eye and head rotations in all dimensions, mice were equipped with two reversibly attached, lateral head-mounted cameras and an inertial measurement unit (IMU) board with an integrated 3-dimensional accelerometer and gyroscope (*Figure 1A and B*; *Video 1*). The estimated error in measurement of head and eye angle were both less than one degree (see Materials and methods). In addition, we recorded the behavior of experimental animals and the cricket prey with an overhead camera to compute the relative position of the mouse and cricket, as well as orientation of the head relative to the cricket. Following our previous studies (*Hoy et al., 2016*; *Hoy et al., 2019*), we defined approaches based on the kinematic criteria that the mouse was oriented towards the cricket and actively moving towards it (see Materials and methods). Together, these recordings and analyses allowed us to synchronously measure eye and head rotations along with cricket and mouse kinematics throughout prey capture behavior (*Figure 1C*; *Video 1*).

The cameras and IMU did not affect overall mouse locomotor speed in the arena or total number of crickets caught per 10 min session (paired t-test, p=0.075; *Figure 1D/E*), suggesting that placement of the cameras and IMU did not significantly impede movement or occlude segments of the visual field required for successful prey capture behavior.

### Eye vergence is stabilized during approach periods

To determine whether mice make convergent eye movements to enhance binocular overlap during approaches, we first characterized the coordination of bilateral eye movements. We defined central eye position, that is 0°, as the average pupil location for each eye, across the recording duration. Measurement of eye position revealed that freely moving mice nearly constantly move their eyes, typically within a ± 20 degree range (*Figures 1C*, *2A and B*), as shown previously (*Meyer et al., 2020*; *Sakatani and Isa, 2007*). *Figure 2C* shows example traces of the horizontal position of the two eyes (top), along with running speed of the mouse (bottom). As described previously (*Wallace et al., 2013*; *Payne and Raymond, 2017*; *Meyer et al., 2018*) and analyzed below (*Figure 3D*), the eyes are generally stable when the mouse is not moving. In addition, the raw traces reveal a pattern of eye movement wherein rapid correlated movements of the two eyes are superimposed on slower anti-correlated movements. The pattern of rapid congruent movements and slower incongruent movements was also reflected in the time-lagged cross-correlation of the change in horizontal position across the two eyes (*Figure 2E*), which was positive at short time lags and negative at longer time lags.

We next calculated the vergence angle, which is the difference in the horizontal position of the two eyes (*Figure 2D*). The range of vergence angles was broadly distributed across negative (converged) and positive (diverged) values during non-approach periods, but became more closely distributed around zero (neutral vergence) during approaches (*Figure 2F*; paired t-test, p=0.024). This can be observed in the individual trace of eye movements before, during, and after an approach (*Figure 2G*, top), showing that while the eyes converge and diverge outside of approach periods, they move in a more coordinated fashion during the approaches. Thus, mice do not converge their eyes nasally to create a wider binocular field during approaches; rather the eyes are more tightly aligned, but at a neutral vergence, during approaches relative to non-approach periods.

Previous studies have demonstrated that eye vergence varies with head pitch (*Wallace et al., 2013*; *Meyer et al., 2018*; *Meyer et al., 2020*). As the head tilts downwards, the eyes move outwards; based on the lateral position of the eyes, this serves to vertically stabilize the visual scene

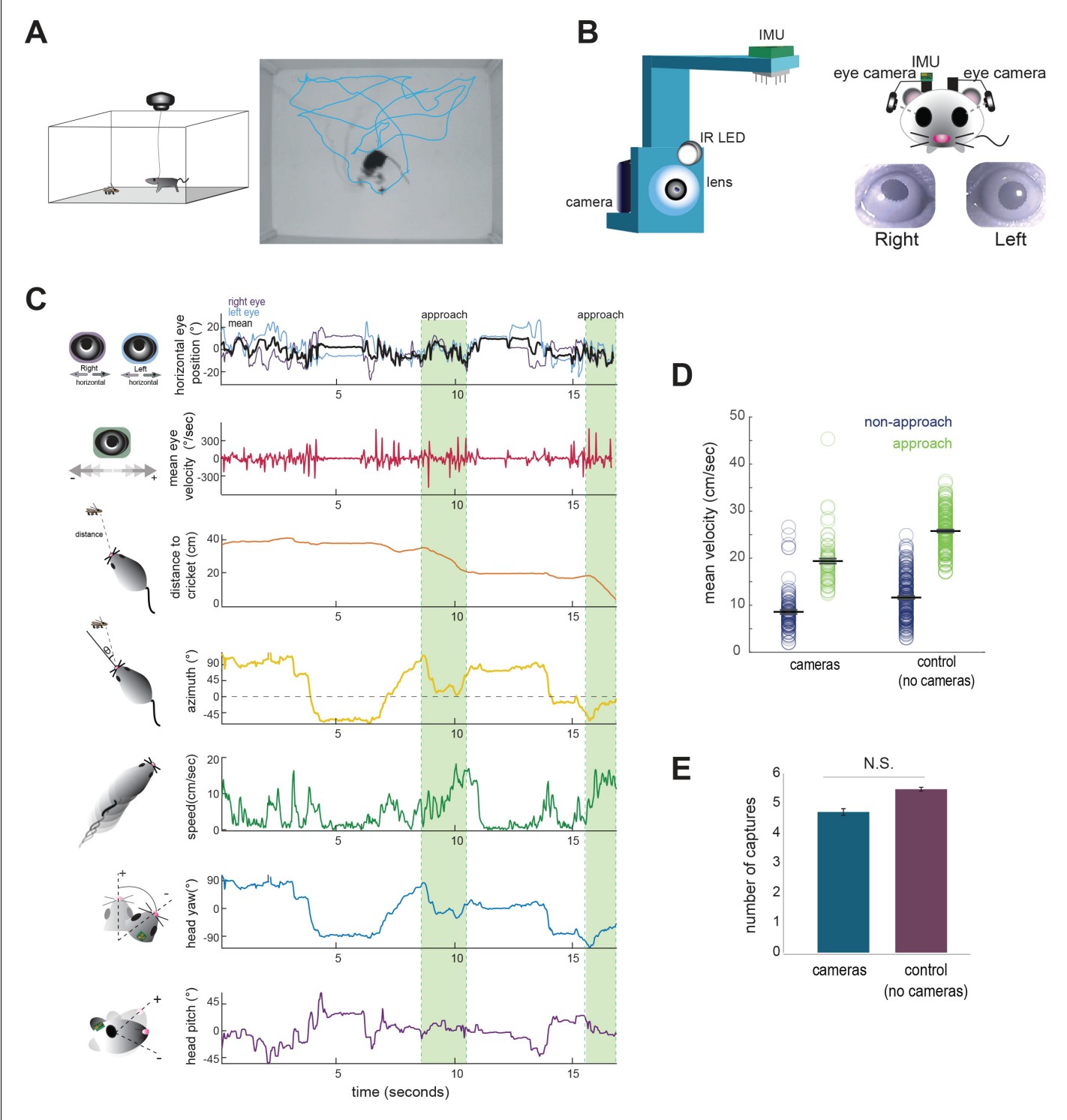

**Figure 1.** Tracking eye and head movements during prey capture. (**A**) Unrestrained mice hunted live crickets in a rectangular plexiglass arena (45 × 38 × 30 cm). Using an overhead camera, we tracked the movement of the mouse and cricket. Example image with overlaid tracks of the mouse (cyan). (**B**) 3D printed holders house a miniature camera, collimating lens, an IR LED, and an IMU, and are reversibly attached to implants on the mouse's head, with one camera aimed at each eye. (**C**) Synchronized recordings of measurements related to bilateral eye position and velocity, mouse position relative to cricket (distance and azimuth, as measured relative to the center of the head), mouse speed, and head rotation in multiple dimensions (analysis here focuses on yaw and pitch). (**D**) Average mouse locomotor speed did not differ across experimental and control experiments (no camera and IMU) for both non-approach and approach periods. Individual dots represent the average velocity per trial. (**E**) Average number of captures per 10 min session

*Figure 1 continued on next page*

*Figure 1 continued*
did not differ between experimental and control sessions (control N = 7 animals, 210 trials; cameras N = 7 animals, 105 trials; two-sample t-test, p=0.075).

relative to changes in head pitch (*Wallace et al., 2013*). We therefore sought to determine whether the stabilization of horizontal eye vergence we observed during approaches reflects corresponding changes in head pitch. Consistent with previous studies, we also found eye vergence to covary with head pitch (*Figure 2H*), such that when the head was vertically centered, the eyes no longer converged or diverged, but were aligned at a neutral vergence (i.e., no difference between the angular positions across the two eyes, see schematic in *Figure 2D*).

Strikingly, we found that while the relationship between head pitch and vergence was maintained during approaches (*Figure 2H*), the distribution of head pitch was more centered during approach periods (*Figure 2H and I*; paired t-test, p=0.0250), indicating a stabilization of the head in the vertical dimension. This can also be seen in the example trace in *Figure 2G*, where the head pitch becomes maintained around zero during approach. These data show that the increased alignment of the two eyes observed during approaches largely represents the stabilization of up/down head rotation, consequently reducing the need for compensatory vergence movements.

## Coordinated horizontal eye movements are primarily compensatory for horizontal head rotations

Next, we aimed to understand the relationship between horizontal head movements (yaw) and horizontal eye movements during approach behavior. In order to isolate the coordinated movement of the two eyes, removing the compensatory changes in vergence described above, we averaged the horizontal position of the two eyes for the remaining analyses (*Figure 3A*). Changes in head yaw and mean horizontal eye position were strongly negatively correlated at zero time lag (*Figure 3B*), suggesting rapid compensation of head movements by eye movements, as expected for VOR-stabilization of the visual scene. The distribution of head and eye movements at zero lag (*Figure 3C*) shows that indeed changes in head yaw were generally accompanied by opposing changes in horizontal eye position, represented by the points along the negative diagonal axis. However, there was also a distinct distribution of off-axis points, representing a proportion of non-compensatory eye movements in which the eyes and head moved in the same direction (*Figure 3C*).

Many studies have reported a limited range of infrequent eye movements in head restrained mice (*Payne and Raymond, 2017*; *Niell and Stryker, 2010*; *Samonds et al., 2018*; *Stahl, 2004*), consistent with the idea that eye movements are generally driven by head movement. Correspondingly in the freely moving context of the prey capture paradigm, we found greatly reduced eye movements when the animals were stationary versus when the animals were running (*Figure 3D*; Kolmogorov-Smirnov test, p=0.032).

We next compared the distribution of mean eye position during approaches and non-approach periods. In contrast to the stabilization of head pitch described above, the distribution of head yaw velocities was not reduced during approaches as shown (*Figure 3E*; paired t-test p=0.938), consistent with the fact that mice must move their heads horizontally as they continuously orient to pursue prey. For both non-approach and approach periods, eye position generally remained within a range less than the size of the binocular zone (±20 degrees; *Figure 3F*, paired t-test, p=0.156), suggesting that the magnitude of eye movements would not shift the binocular zone to an entirely new location. Comparison of horizontal eye velocity

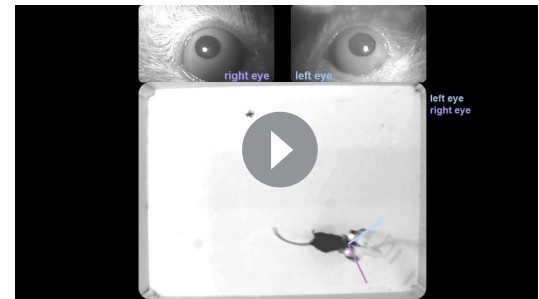

**Video 1.** Video of mouse performing prey capture with reversibly attached eye cameras, demonstrating synchronized measurement of bilateral eye positions and mouse/cricket behavior. The direction of each eye is superimposed on the head (purple and light blue lines) based on calculated eye position and head angle.
https://elifesciences.org/articles/57458#video1

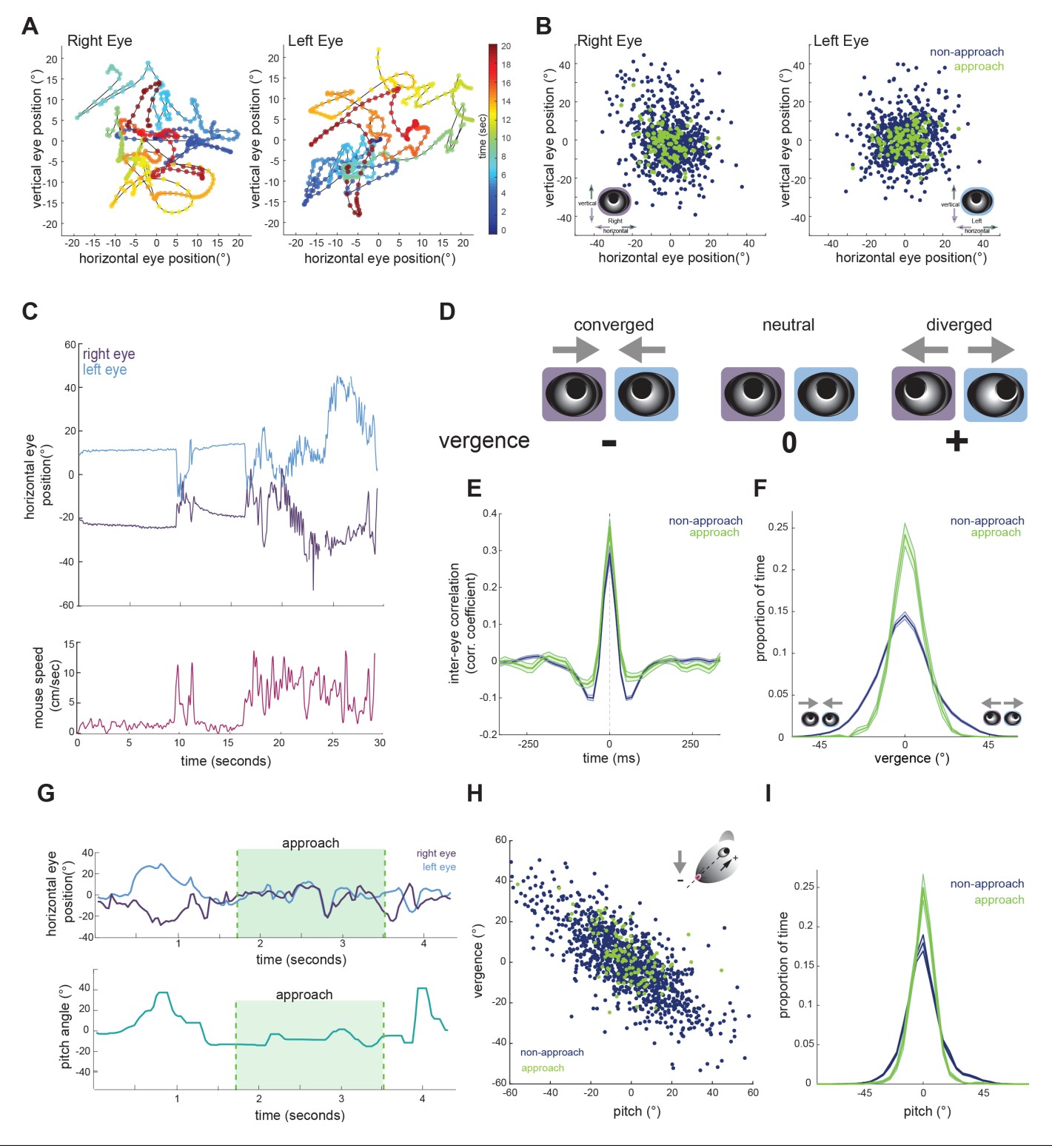

**Figure 2.** Eye position is more aligned across the two eyes during approach periods. (**A**) Example eye movement trajectory for right and left eyes for a 20 s segment, with points color-coded for time. (**B**) Horizontal and vertical position for right and left eyes during approach and non-approach times. N = 7 animals, 105 trials, 792 time pts (non-approach), 110 time pts (approach), representing a random sample of 0.22% of non-approach and 0.52% of approach time points. (**C**) Example trace of horizontal eye positions (top) and running speed (bottom) for a 30 s segment. (**D**) Schematic demonstrating vergence eye movements. (**E**) Cross correlation of horizontal eye position across the two eyes for non-approach and approach periods. (**F**) Histogram of vergence during non-approach and approach. (**G**) Example trace of horizontal eye position (top) and head pitch (bottom) before, during, and after an

*Figure 2 continued on next page*

*Figure 2 continued*

approach. (H) Scatter plot of head pitch and eye vergence. As head pitch tilts downwards, the eyes move temporally to compensate (as in schematic). N = 7 animals, 105 trials, 1240 time points (non-approach), 132 time points (approach), representing a sample of 0.35% of non-approach and 0.63% of approach time points. (I) Histogram of head pitch during approach and non-approach periods, across all experiments.

between non-approach and approach epochs revealed that the eyes move with similar dynamics across both behavioral periods (*Figure 3G*, panel 1; paired t-test, p=0.155). Additionally, at times when head yaw was not changing, horizontal eye position also did not change (*Figure 3G*, panel 2; paired t-test, p=0.229). Together, these observations suggest that most coordinated eye movements in the horizontal axis correspond to changes in head yaw, and that the eyes do not scan the visual environment independent of head movements or when stationary.

## Non-compensatory saccades shift gaze position

Gaze position - the location the eyes are looking in the world - is a combination of the position of the eyes and the orientation of the head. Compensatory eye movements serve to prevent a shift in gaze, whereas non-compensatory eye movements (i.e., saccades) shift gaze to a new position.

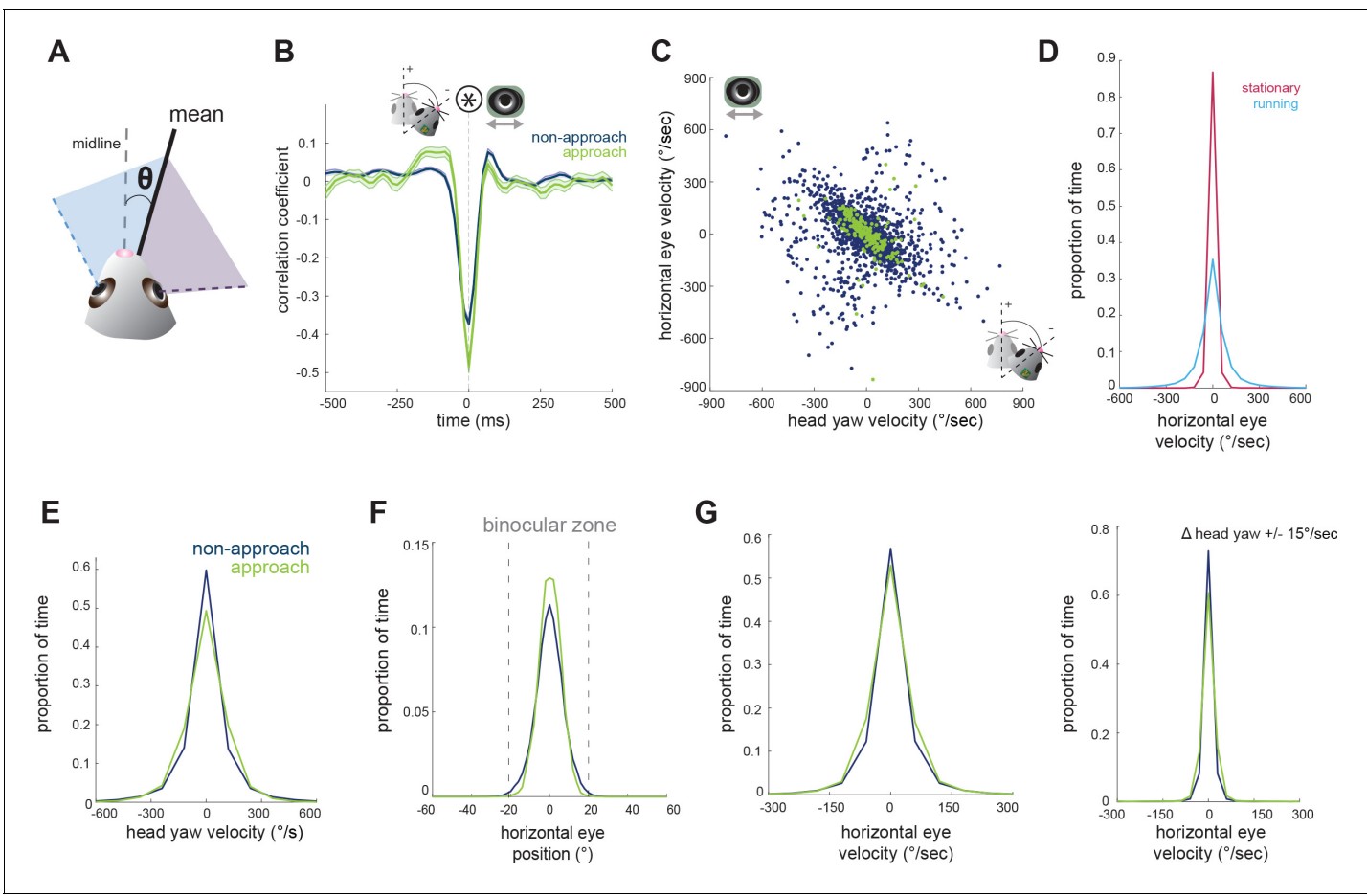

**Figure 3.** Horizontal eye movements are mostly compensatory for yaw head rotations. (A) To remove the effect of non-conjugate changes in eye position (i.e. vergence shifts), we compute the average angular position of the two eyes. (B) Cross-correlation of change in head yaw and horizontal eye position. (C) Scatter plot of horizontal rotational head velocity and horizontal eye velocity. N = 7 animals, 105 trials, 3565 (non-approach) and 211 (approach) timepoints, representing 1% of non-approach and 1% of approach timepoints. (D) Distribution of horizontal eye position during stationary and running periods (defined as times when mouse speed is greater than 1 cm/sec; Kolmogorov-Smirnov test, p=0.032). (E) Distribution of head angle velocity (paired t-test, p=0.938). (F) Distribution of mean absolute eye position (paired t-test, p=0.156). (G) Distribution of horizontal eye velocity (paired t-test, p=0.155) and distribution of eye velocity when head yaw is not changing (change in head yaw between ±15 deg/sec; paired t-test, p=0.229; N = 7 animals, 105 trials).

Although the vast majority of eye movements are compensatory for head movements, as demonstrated by strong negative correlation in *Figure 3B/C*, a significant number of movements are not compensatory, as seen by the distribution of off-axis points in *Figure 3C*. These eye movements will therefore shift the direction of the animal's gaze relative to the environment. We next examined how eye movements, and particularly non-compensatory movements, contribute to the direction of gaze during free exploration and prey capture. In particular, are these gaze shifts directed at the target prey?

We segregated eye movements into compensatory versus gaze-shifting by setting a fixed gaze velocity threshold of ±180 °/sec, based on the gaze velocity distribution (*Figure 4A*), which shows a transition between a large distribution around zero (stabilized gaze) and a long tail of higher velocities (rapid gaze shifts). This also provides a clear segregation in the joint distribution of eye and head velocity (*Figure 4B*), with a large number of compensatory gaze-stabilizing movements (black points) where eye and head motion are anti-correlated, and much smaller population of gaze shifts (red). This classification approach provides an alternative to standard primate saccade detection (*Andersson et al., 2017*; *Stahl, 2004*; *Matthis et al., 2018*), which is often based on eye velocity rather than gaze velocity, since in the freely moving condition, particularly in afoveate species, rapid gaze shifts (saccades) often result from a combination of head and non-compensatory eye movements, rather than eye movements alone (*Land, 2006*).

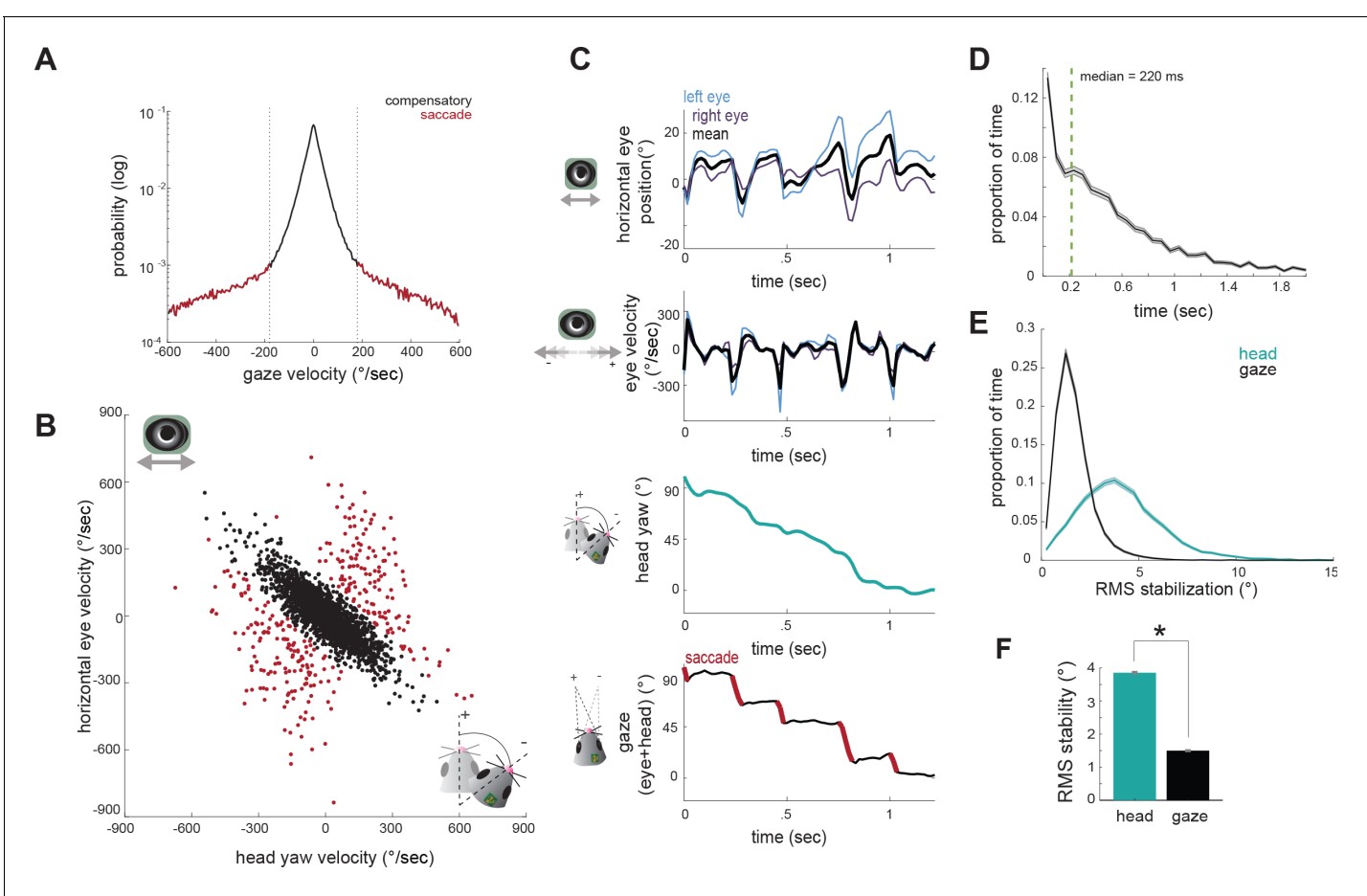

**Figure 4.** Compensatory and non-compensatory eye movements generate a saccade-and-fixate gaze pattern during head turns. (**A**) Distribution of gaze velocity (N = 377459 time points) showing segregation of non-compensatory and compensatory movements with thresholds at ±180°/sec. (**B**) Joint distributions of head yaw and horizontal eye velocity colored by their type as defined in A. Black points represent compensatory movements and red represent non-compensatory saccadic movements. Points shown are a random sample of 2105 approach timepoints, 10% of total approach time points. (**C**) Example traces of horizontal eye position, head yaw, and gaze demonstrate a saccade-and-fixate pattern in gaze. (**D**) Histogram of fixation duration; fixations N = 9730, 105 trials. (**E**) Root Mean Squared (RMS) stabilization histograms for head yaw and gaze. (**F**) Bar graphs are medians of RMS stabilization distributions (median head = 3.87 deg; median gaze = 1.5 deg; paired t-test, p=0).

We next determined how compensatory and non-compensatory eye movements contribute to the dynamics of gaze during ongoing behavior, by computing the direction of gaze as the sum of eye position and head position. Strikingly, the combination of compensatory and non-compensatory eye movements (*Figure 4C*, top) with continuous change in head orientation (*Figure 4C*, middle) results in a series of stable gaze positions interspersed with abrupt shifts (*Figure 4C*, bottom). This pattern of gaze stabilization interspersed with rapid gaze-shifting movements, known as 'saccade-and-fixate,' is present across the animal kingdom and likely represents a fundamental mechanism to facilitate visual processing during movement (*Land, 1999*). These results demonstrate that the mouse oculomotor system also engages this fundamental mechanism.

Durations of fixations between saccades showed wide variation, with a median of 220 ms (*Figure 4D*). To quantify the degree of stabilization achieved, we compared the root mean square (RMS) deviation of gaze position and head yaw during stabilization periods (*Figure 4E*). This revealed that the gaze is nearly three times less variable than the head (*Figure 4F*; median head = 3.87 deg; median gaze = 1.58 deg; p=0), resulting in stabilization to within nearly one degree over extended periods, even during active approach toward the cricket.

## Targeting of gaze relative to cricket during approach

Saccade-and-fixate serves as an oculomotor strategy to sample and stabilize the visual world during free movement. In primates, saccades are directed towards specific targets of interest in the visual field. Is this true of the non-compensatory movements in the mouse? In other words, do saccades directly target the cricket? To address this, we next analyzed the dynamics of head and gaze movements relative to the cricket position during hunting periods, to compare how accurately the direction of the gaze and the head targeted the cricket during saccades.

*Figure 5A* shows example traces of head and eye dynamics across an approach period (see also *Video 2*). Immediately before approaching the cricket, the animal begins a large head turn towards the target, thereby reducing the azimuth angle (center of the head relative to cricket). This head turn is accompanied by a non-compensatory eye movement in the same direction (*Figure 5A*, 3rd panel, see mean trace in black) that accelerates the shift in gaze. Then during the approach, the eyes convert the continuous tracking of the head into a series of stable locations of the gaze (black sections in *Figure 5A*, bottom). Note also the locking of the relative position of the two eyes (*Figure 5A*, 3rd panel, blue and purple), as described above in *Figure 2*.

To determine how head and eye movements target the prey, we computed absolute value traces of head and gaze angle relative to cricket (head and gaze azimuth), and aligned these to the onset of each non-compensatory saccadic eye movement. The average of all traces during approaches revealed that saccades are associated with a head turn towards the cricket, as shown by a decrease in the azimuth angle (*Figure 5B*). Immediately preceding a saccade, the gaze is stabilized while the head turns, and the saccade then abruptly shifts the gaze. Notably, following the saccade, the azimuth of gaze is the same as the azimuth of the head, suggesting that eye movements are not targeting the cricket more precisely, but simply 'catching up' with the head, by re-centering following a period of stabilization.

To further quantify this, we assessed the accuracy of the head and gaze at targeting cricket position before and after saccades. Preceding saccades, the distribution of head angles was centered around the cricket, while the gaze less accurately targeted and was offset from the cricket to the left or right (*Figure 5C*/5D top; paired t-test; p=8.48×$10^{-9}$ p=2×$10^{-5}$), due to compensatory stabilization. After the saccade, however, gaze and head were equally targeted towards the cricket (*Figure 5C/D* bottom; p=0.979p=0.4), as the saccade recentered the eyes relative to the head and thereby the cricket. This pattern of stabilizing the gaze and then saccading to recenter the gaze repeats whenever the head turns until capture is successful (see *Video 2*).

Further supporting a strategy where the head guides targeting, with the eyes following to compensate, we examined how both head and eye movements are correlated with the cricket's position. At short latencies, the change in head angle relative to the location of the cricket was highly correlated (*Figure 5E*), indicating that during approach the animal is rapidly reorienting its head towards the cricket. However, the change in gaze with the azimuth instead showed only a weak correlation because the eyes themselves are not always aligned with the azimuth due to stabilization periods (*Figure 5F*). Together, these results suggest that in mice, tracking of visual objects in freely moving

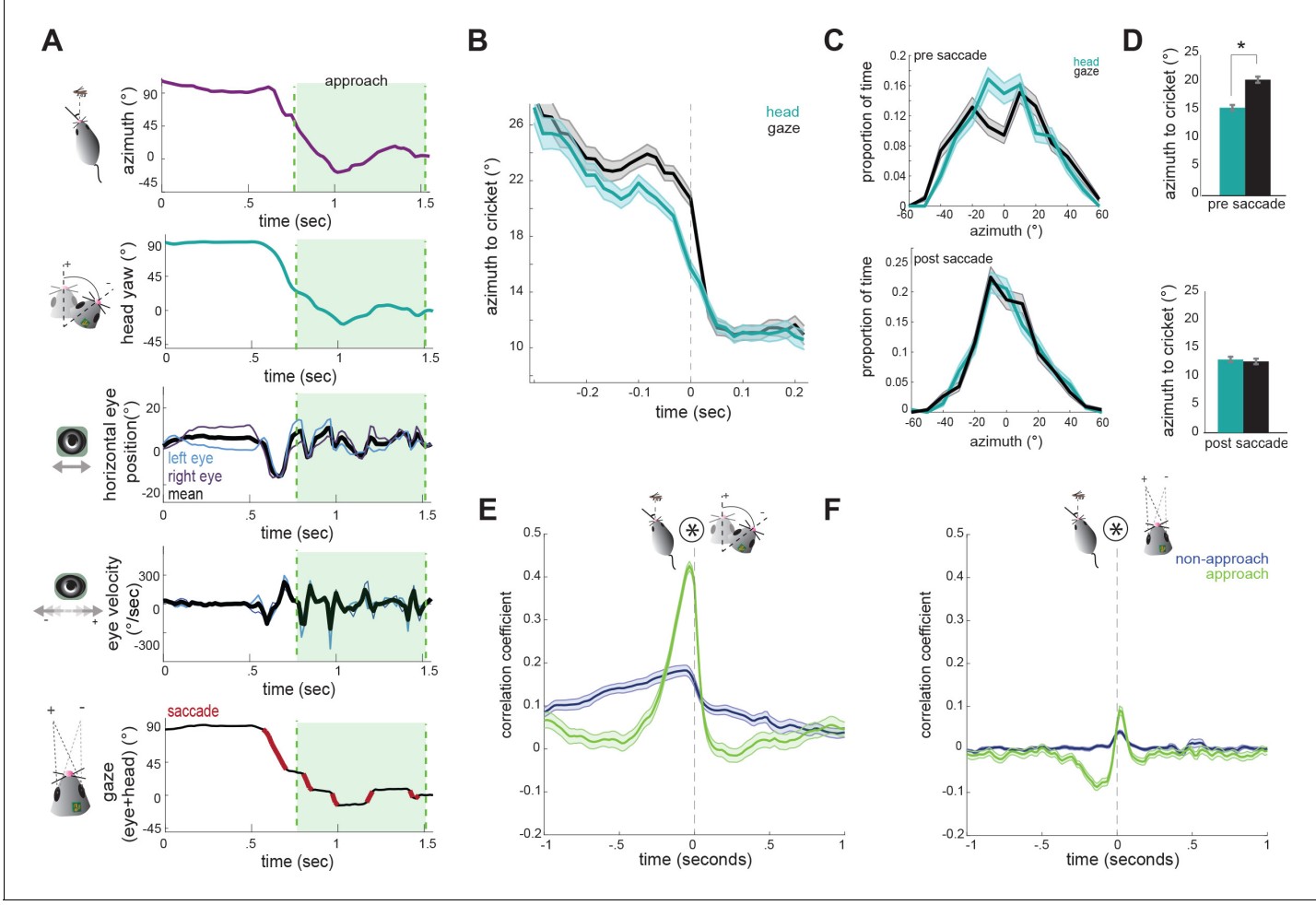

**Figure 5.** Head angle tracks cricket position more accurately than gaze position. (A) Example traces of horizontal eye position, azimuth to cricket, head yaw, and gaze demonstrate a saccade-and-fixate pattern in gaze before and during an approach period. The head is pointed directly at the cricket when azimuth is 0˚. Note the rapid decrease in azimuth, head yaw, and mean horizontal eye position creating a saccade immediately preceding the start of approach. (B) Average head yaw and gaze around the time of saccade as a function of azimuth to the cricket. Time = 0 is the saccade onset. (C) Histograms of head yaw and gaze position before and after saccades occur. (D) Medians of yaw and gaze distributions from C (paired t-test, $p_{pre saccade}=8.48 \times 10^{-9}$; $p_{post saccade}=0.979$). (E) Cross correlation of azimuth and change in head yaw for non-approach and approach periods. (F) Cross correlation of azimuth and change in gaze for non-approach and approach periods. N = 105 trials, 7 animals.

contexts is mediated through directed head movements, and corresponding eye movements that stabilize the gaze and periodically update to recenter with the head as it turns.

## Discussion

Here we investigated the coordination of eye and head movements in mice during a visually guided ethological behavior, prey capture, that requires the localization of a specific point in the visual field. This work demonstrates that general principles of coordinated eye and head movements, observed across species, are present in the mouse. Additionally, we address the potential targeting of eye movements towards behaviorally relevant visual stimuli, specifically the moving cricket prey. We find that tracking is achieved through directed head movements that accurately target the cricket prey, rather than directed, independent eye movements. Together, these findings define how mice move

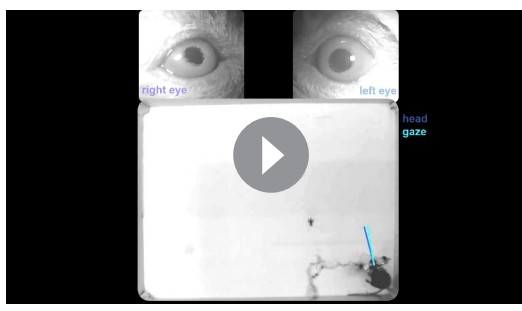

**Video 2.** Video of mouse performing prey capture, demonstrating dynamics of head orienting (dark blue) and gaze direction (cyan). Note that during head turns the gaze is transiently offset from the head angle vector, due to compensatory eye movements, creating a stable image for the animal. Then, non-compensatory saccades shift the gaze position such that it aligns with the head to accurately target the cricket.
https://elifesciences.org/articles/57458#video2

their eyes to achieve an ethological behavior and provide a foundation for studying active visually-guided behaviors in the mouse.

One potential limitation of our eye tracking system is the 60 Hz framerate of the miniature cameras. This temporal resolution is significantly lower than traditional eye tracking paradigms using videography or eye-coil systems in head-restrained humans, non-human primates, and rodents (*Payne and Raymond, 2017*; *Sakatani and Isa, 2007*), though similar to recent video-based tracking in freely moving rodents (*Meyer et al., 2018*; *Meyer et al., 2020*) and humans (*Matthis et al., 2018*). We do not expect that this would significantly alter our findings, as the basic parameters of eye movements (amplitude and speed) that we found (*Figures 2B, 3C and F*) were similar to measurements made in both head-fixed mice with high-speed videography (*Sakatani and Isa, 2007*) and freely moving mice with a magnetic sensor (*Payne and Raymond, 2017*). However, although we are able to detect peak velocities over 300°/sec, we may still be under-estimating the peak velocity during saccades. Therefore increasing the temporal resolution further could lead to more robust detection of rapid gaze shifts and would potentially enhance classification of saccadic eye movements.

We found a pattern of gaze stabilization interspersed with abrupt, gaze-shifting saccades during both non-approach and approach epochs. This oculomotor strategy has been termed 'saccade-and-fixate' (reviewed in *Land, 1999*), and is present in most visual animal species, from insects to primates, and was recently demonstrated in mice (*Meyer et al., 2020*). In primates, gaze shifts can be purely driven by eye movements, but in other species saccades generally correspond to non-compensatory eye movements during head rotation, suggesting transient disengagement of VOR mechanisms. These saccadic movements are present in invertebrates and both foveate and non-foveate vertebrates (reviewed in *Land, 1999*), and work to both recenter the eyes and relocate the position of gaze as animals turn. We found that these brief congruent head and eye movements are interspersed with longer duration (median ~200 ms) periods of compensatory movements, which stabilize the gaze to within nearly 1° as the head continues to rotate. Together these eye movements function to create a stable series of images on the retina even during rapid tracking of prey.

However, the saccade-and-fixate strategy raises the question of whether mice actively target a specific relevant location with saccadic eye movements. We examined this during periods of active approach toward the cricket to determine whether the eyes specifically target the cricket, relative to head orientation. During approaches, most saccades occur during corrective head turns toward the cricket location. While saccades do bring the gaze closer to the cricket, they do not do so more accurately than the head direction. In fact, prior to the saccade, mice sacrifice centering of the gaze on the target to instead achieve visual scene stability. The eyes then 'catch up' to the head as it is rotating (*Figure 5B/C*). Thus, these eye movements serve to reset eye position with the head, rather than targeting the cricket specifically. Combined with the fact that mice do not make significant eye movements in the absence of head movements (*Figure 3F*), this suggests that mice do not perform either directed eye saccades or smooth pursuit, which are prominent features of primate vision. On the other hand, the fact that mice use a saccade-and-fixate strategy makes it clear that they are still actively controlling their retinal input, despite low visual acuity. Indeed, the saccade-and-fixate strategy makes mouse vision consistent with the vast majority of species across the animal kingdom.

We also examined whether mice make specific vergence eye movements that could serve to modulate the binocular zone, as in some other species with eyes located laterally on the head. We find that rather than moving the eyes nasally to expand the binocular zone, during approach toward the cricket the two eyes become stably aligned, but at a neutral vergence angle that is neither converged or diverged (*Figure 2E*). While several species with laterally-placed eyes use convergent eye

movements during prey capture to create a wider binocular field (*Feord et al., 2020*; *Bianco et al., 2011*; *Martin, 2009*; *Katz et al., 2015*), our results show that mice do not utilize this strategy during prey capture. However, vergence eye movements in rodents have previously been shown to compensate for head tilt (*Wallace et al., 2013*), and correspondingly we find that during approach periods mice stabilize head tilt. Thus, the stable relative alignment of the two eyes during approach likely reflects stabilization of the head itself. These results suggest that the 40 degree binocular zone is sufficient for tracking centrally located objects, as the eyes to not move to expand this during approaches. This is consistent with previous work showing that during active approach the mouse's head is oriented within ±15 degrees relative to the cricket (*Hoy et al., 2016*), meaning that even the resting binocular zone would encompass the cricket. However, it remains to be determined whether mice actually use binocular disparity for depth estimation during prey capture. A recent study demonstrated that mouse V1 encodes binocular disparities spanning a range of 3–25 cm from the mouse's head (*Land, 2018*), suggesting that disparity cues are available at the typical distances during approach (interquartile range 14.6 cm to 27.6 cm). Alternatively, mice may use retinal image size or other distance cues, or may simply orient to the azimuthal position of the cricket regardless of distance.

The finding that mice do not specifically move their eyes to target a location does not preclude the possibility that different regions of retinal space are specialized for certain processing. In fact, as a result of targeting head movements, the cricket prey is generally within the binocular zone during approach, so any mechanisms of enhanced processing in the binocular zone or lateral retina would still be behaviorally relevant. Anatomically, there is a gradient in density of different retinal ganglion cell types from medial to lateral retina (*Bleckert et al., 2014*). Likewise behavioral studies have shown enhanced contrast detection when visual stimuli are located in the binocular field, rather than the monocular fields (*Speed et al., 2019*). Based on the results presented here, in mice these specializations are likely to be engaged by head movements that localize stimuli in the binocular zone in front of the head, as opposed to primates, which make directed eye movements to localize stimuli on the fovea.

Together, the present findings suggest that orienting relative to visual cues is driven by head movements rather than eye movements in the mouse. This is consistent with the general finding that for animals with small heads it is more efficient to move the head, whereas animals with large heads have adapted eye movements for rapid shifts to overcome the inertia of the head (*Land, 2018*). From the experimental perspective, this suggests that head angle alone is an appropriate measure to determine which visual cues are important during study of visually guided, goal-directed behaviors in the mouse. However, measurements of eye movements will be essential for computing the precise visual input animals receive (i.e., the retinal image) during ongoing freely moving behaviors, and how this visual input is processed within visual areas of the brain.

The saccade-and-fixate strategy generates a series of stable visual images separated by abrupt changes in gaze that shift the visual scene and location of objects on the retina. How then are these images, interleaved with periods of motion blur, converted into a continuous coherent percept that allows successful natural behaviors to occur? Anticipatory shifts in receptive field location during saccades, as well as gaze position-tuned neural populations, have been proposed as mechanisms in primates to maintain coherent percepts during saccades, while corollary discharge, saccadic suppression, and visual masking have been proposed to inhibit perception of motion blur during rapid eye movements (*Higgins and Rayner, 2015*; *Wurtz, 2008*). However, the mechanisms that might mediate these, at the level of specific cell types and neural circuits, are poorly understood. Studying these processes in the mouse will allow for investigation of the neural circuit basis of these perceptual mechanisms through the application of genetic tools and other circuit dissection approaches (*Huberman and Niell, 2011*; *Luo et al., 2008*). Importantly, most of our visual perception occurs during active exploration of the environment, where the combined dynamics of head and eye movements create a dramatically different image processing challenge than typical studies in stationary subjects viewing stimuli on a computer monitor. Examination of these neural mechanisms will extend our understanding of how the brain performs sensory processing in real-world conditions.

# Materials and methods

**Key resources table**

| Reagent type (species) or resource | Designation | Source or reference | Identifiers | Additional information |
|---|---|---|---|---|
| Strain, strain background (*Mus musculus*) | C57Bl/6J | JAX | JAX: 000664 | Wild type animals |
| Software, algorithm | Matlab | Matlab | Matlab R2020a | |
| Software, algorithm | DeepLabCut | *Mathis et al., 2018* | | |
| Software, algorithm | Bonsai | *Lopes et al., 2015* | | |

## Animals

All procedures were conducted in accordance with the guidelines of the National Institutes of Health and were approved by the University of Oregon Institutional Animal Care and Use Committee (protocol number 17–27). Animals used for this study were wild-type (C57 Bl/6J) males and females (3 males and four females) aged 2–6 months.

## Prey capture behavior

Prey capture experiments were performed following the general paradigm of *Hoy et al., 2016*. Mice readily catch crickets in the homecage without any training or habituation, even on the first exposure to crickets. However, we perform a standard habituation process to acclimate the mice to being handled by the experimenters, hunting within the experimental arena, and wearing cameras and an IMU while hunting. Following six 3 min sessions (over 1–2 days) of handling, the animals were placed in the prey capture arena to explore with their cagemates. The duration of this group habituation was at least six 10 min sessions over 1–2 days. One cricket (Rainbow mealworms, 5 week old) per mouse was placed in the arena with the mice for the last half of the habituation sessions. For the subsequent habituation step, the mice were placed in the arena alone with one cricket for 7–10 min. This step was repeated for 2–3 training days (6–9 sessions) until most mice successfully caught crickets within the 10 min period.

Animals were then habituated to head-fixation above a spherical Styrofoam treadmill (*Dombeck et al., 2007*). Head fixation was only used to fit, calibrate, and attach cameras before experiments. Cameras were then fitted to each mouse (described below) and mice were habituated to wearing the cameras while walking freely in the arena, which took 1–2 sessions lasting 10 min. After the animals were comfortable with free locomotion with cameras, they were habituated to hunting with cameras attached. This took roughly one to two e hunting sessions of 10 min duration for each mouse. The animals were then food deprived for a period of ~12–18 hr and then run in the prey capture assay for three 10 min sessions per data collection day. Although animals will hunt crickets without food restriction, this allowed for more trials within a defined experimental period.

The rectangular prey capture arena was a white arena of dimensions 38 × 45×30 cm (*Hoy et al., 2016*). The arena was illuminated with a 15 Watt, 100 lumen incandescent light bulb placed roughly one meter above the center of the arena to mimic lux during dawn and dusk, times at which mice naturally hunt (*Bailey and Sperry, 1929*). Video signal was recorded from above the arena using a CMOS camera (Basler Ace, acA2000–165 umNIR, 30 Hz acquisition).

Following the habituation process, cameras were attached and mice were placed in the prey capture arena with one cricket. Experimental animals captured and consumed the cricket before a new cricket was placed in the arena. The experimenters removed any residual cricket pieces in the arena before the addition of the next cricket. A typical mouse catches and consumes between 3–5 crickets per 10 min session. Control experiments were performed using the same methods, but with no cameras or IMU attached.

## Surgical procedure

To allow for head-fixation during initial eye camera alignment, before the habituation process mice were surgically implanted with a steel headplate, following *Niell and Stryker, 2010*. Animals were anesthetized with isoflurane (3% induction, 1.5–2% maintenance, in $O_2$) and body temperature was maintained at 37.5°C using a feedback-controlled heating pad. Fascia was cleared from the surface of the skull following scalp incision and a custom steel headplate was attached to the skull using Vet-bond (3M) and dental acrylic. The headplate was placed near the back of the skull, roughly 1 mm anterior of Lambda. A flat layer of dental acrylic was placed in front of the headplate to allow for attachment of the camera connectors. Carprofen (10 mg/kg) and lactated Ringer's solution were administered subcutaneously and animals were monitored for three days following surgery.

## Camera assembly and head-mounting

To measure eye position, we used miniature cameras that could be reversibly attached to the mouse's head via a chronically implanted Millmax connector. The cameras (1000 TVL Mini CCTV Camera; iSecurity101) were 5 × 6 × 6mm with a resolution of 480 × 640 pixels and a 78 degree viewing angle, and images were acquired at 30 Hz. Some of the cameras were supplied with a built in NIR blocking filter. For these cameras, the lens was unscrewed and the glass IR filter removed with fine forceps. A 200 Ohm resistor and 3 mm IR LED were integrated onto the cameras for uniform illumination of the eyes. Power, ground, and video cables were soldered with lightweight 36 gauge FEP hookup wire (Cooner Wire; CZ 1174). A 6 mm diameter collimating lens with a focal distance of 12 mm (Lilly Electronics) was inserted into custom 3D printed housing and the cameras were then inserted and glued behind this (see *Figure 1* for schematic of design). The inner side of the arm of the camera holder housed a male Mill-Max connector (Mill-Max Manufacturing Corp. 853-93-100-10-001000) cut to 5 mm (2 rows of 4 columns), used for reversible attachment of the cameras to the implants of experimental animals. A custom IMU board with integrated 3-dimensional accelerometer and gyroscopes (Rosco Technologies) was attached to the top of one of the camera holders (see *Figure 1B*). The total weight of the two cameras together, with the lenses, connectors, 3D printed holders, and IMU was 2.6 grams. Camera assemblies were fitted onto the head by attaching them to corresponding female Mill-Max connectors. Cameras were located in the far lateral periphery of the mouse's visual field, roughly 100° lateral of the head midline and 40 degrees above the horizontal axis, and covered roughly 25 × 25° of the visual field. When the camera was appropriately focused on the eye, the female connectors were glued onto the acrylic implant using cyanoacrylate adhesive (Loctite). Because the connectors were each positioned during this initial procedure and permanently fixed in place, no adjustment of camera alignment was needed for subsequent experimental days. With this system, the average magnitude of camera shake jitter across experiments was 0.49 + / - 0.33 pixels (mean + / - s.d., N = 7 animals), as measured by computing the RMS frame-to-frame jitter of stationary points on the animal's head (base of the implant) in the recorded videos.

## Mouse and cricket tracking

Video data with timestamps for the overhead camera were acquired at 30 frames per second (fps) using Bonsai (*Lopes et al., 2015*). We used DeepLabCut (*Mathis et al., 2018*) for markerless estimation of mouse and cricket position from overhead videos. For network training, we selected eight points on the mouse head (nose, two camera connectors, two IR LEDs, two ears, and center of the head between the two ears), and two points for the cricket (head and body). Following estimation of the selected points, analysis was performed with custom MATLAB scripts, available at *Michaiel et al., 2020*.

To determine periods when the animal was moving versus stationary, head movement speed was median filtered across a window of 500 ms and a threshold of 1 cm/sec was applied. Position and angle of the head were computed by fitting the eight measured points on the head for each video frame to a defined mean geometry plus an x-y translation and horizontal rotation. The head direction was defined as the angle of this rotation, referenced to the line between the nose and center of the head. We also used this head-centered reference to compute the azimuth, which is the angle of the mouse relative to the cricket. Following *Hoy et al., 2016*, we defined approaches as times at which the velocity of the mouse was greater than 1 cm/sec, the azimuth of the mouse was between −45

and 45 degrees relative to cricket location, and the distance to the cricket was decreasing at a rate greater than 10 cm/sec. Although mice eventually catch and consume the cricket in each trial, and are motivated to hunt due to food restriction, we cannot rule out the possibility that some approach periods may represent tracking or chasing without the intent to capture.

Analog voltage signals from the IMU were recorded using a LabJack U6 at 50 Hz sampling rate. Voltages from the accelerometer channels were median filtered with a window of 266.7 ms to remove rapid transients and converted to m/sec$^2$, providing angular head orientation. Voltages from the gyroscope channels were converted to radians/sec without filtering, providing head rotation velocity.

## Eye tracking and eye camera calibration

Video data with timestamps for the two eyes were acquired at 30fps using Bonsai. The video data are delivered by the camera in NTSC format, an interlaced video format in which two sequential images (acquired at 60fps) are interdigitated into each frame on alternate horizontal lines. We therefore de-interlaced the video in order to restore the native 60fps resolution by separating out alternate lines of each image. We then linearly downsampled the resolution along the horizontal axis by a factor of two, to match spatial resolution in horizontal and vertical dimensions.

To track eye position, we used DeepLabCut (*Mathis et al., 2018*) to track eight points along the edge of the pupil. The eight points were then fit to an ellipse using the least-squares criterion. In order to convert pupil displacement into angular rotation, which cannot be calibrated by directed fixation as in primates, we followed the methods used in *Wallace et al., 2013*. This approach is based on the principle that when the eye is pointed directly at the camera axis, the pupil is circular, and as the eye rotates, the circular shape flattens into an ellipse depending on the direction and degree of angular rotation from the center of the camera axis. To calculate the transformation of a circle along the camera axis to the ellipse fit, two pieces of information are needed: the camera axis center position and the scale factor relating pixels of displacement to angular rotation. To find the camera axis, we used the constraint that the major axis of the pupil ellipse is perpendicular to the vector from the pupil center to the camera axis center. This defines a set of linear equations for all of the pupil observations with significant ellipticity, which are solved directly with a least-squares solution. Next, the scale factor was estimated based on the equation defining how the ellipticity of the pupil changes with the corresponding shift from the camera center in each video frame. Based on the camera center and scale factor for each video, we calculated the affine transformation needed to transform the circle to the ellipse fit of the pupil in each frame, and the angular displacement from the camera axis was then used for subsequent analyses. Mathematical details of this method are presented in *Wallace et al., 2013*.

Following computation of kinematic variables (mouse, cricket, and eye position/rotation), these values were linearly interpolated to a standard 60 Hz timestamp to account for differences in acquisition timing across the multiple cameras and the IMU.

To characterize the robustness of the tracking system, we estimated the error in eye and head position measurements. As there is no ground truth measurement for eye position to compare to, we estimated an upper bound based on the stability of the eye when the head was stationary, as this is when eye movements are expected to be minimal. Specifically, we computed the standard deviation of horizontal eye position between frames during times when mouse speed was less than 1 cm/sec and head rotation <1 degree. We computed the error to be 0.51 + / - 0.25 degrees (mean + / - s.d., n = 105 trials). Similarly, to estimate the error of head angle measurements, we compared the independent estimates of head yaw rotation between frames as measured by both the IMU and DeepLabCut tracking. These measures have an RMS difference of 0.95 + / - 0.25 (mean+/-s.d., n = 105 trials), which represents an upper bound as it is based on the combined error of these two measurements separately. Thus, we infer that errors in estimating eye and head position are both less than one degree.

## Quantification and statistical analyses

Two-tailed paired t-tests or Wilcoxon Rank sum tests were used to compare data between non-approach and approach epochs. For comparisons between experimental and control groups, two-sample tests (Kolmogorov-Smirnov or two-sample two-tailed t-test) were used. Significance was

defined as p<0.05, although p-values are presented throughout. In all figures, error bars represent ±the standard error of the mean or median, as appropriate.

## Acknowledgements

We thank Yichen Fan and Alyssa Fuentez for their help with behavioral data collection; the Wehr lab for assistance in implementing head-mounted cameras; members of the Niell lab for helpful discussions; and Dr. Marina Garrett, Dr. Clifford Keller, Dr. Jude Mitchell, and Dr. Matthew Smear for feedback on the manuscript. This work was supported by NIH R34NS111669 (CMN) and the University of Oregon Promising Scholar Award (AMM).

## Additional information

### Funding

| Funder | Grant reference number | Author |
| --- | --- | --- |
| National Institutes of Health | R34NS111669 | Cristopher M Niell |
| University of Oregon | Promising Scholar Award | Angie M Michaiel |

The funders had no role in study design, data collection and interpretation, or the decision to submit the work for publication.

### Author contributions

Angie M Michaiel, Conceived the project, Developed methodology and designed experiments, Analyzed data, Wrote the manuscript, Created figures; Elliott TT Abe, Wrote camera calibration software, Contributed to data analysis, Manuscript preparation; Cristopher M Niell, Conceived the project, Designed experiments, Analyzed data, Wrote manuscript, Provided resources

### Author ORCIDs

Angie M Michaiel (iD) https://orcid.org/0000-0002-5312-8329
Cristopher M Niell (iD) https://orcid.org/0000-0001-6283-3540

### Ethics

Animal experimentation: All procedures were conducted in accordance with the guidelines of the National Institutes of Health and were approved by the University of Oregon Institutional Animal Care and Use Committee (Protocol number: 17-27).

### Decision letter and Author response

Decision letter https://doi.org/10.7554/eLife.57458.sa1
Author response https://doi.org/10.7554/eLife.57458.sa2

## Additional files

### Supplementary files

• Transparent reporting form

### Data availability

Behavioral data has been submitted to Dryad with DOI https://doi.org/10.5061/dryad.8cz8w9gmw.

The following dataset was generated:

| Author(s) | Year | Dataset title | Dataset URL | Database and Identifier |
| --- | --- | --- | --- | --- |
| Michaiel AM, Abe ETT, Niell CM | 2020 | Data from: Dynamics of gaze control during prey capture in | https://doi.org/10.5061/dryad.8cz8w9gmw | Dryad Digital Repository, 10.5061/ |

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
