## [Decision Letter]

**Acceptance summary:**

This paper provides novel insights into how mice actively acquire visual information in natural contexts by moving their eyes and head. Findings provide a foundation for studying active vision and movement during ethological behaviors such as prey catching.

**Decision letter after peer review:**

Thank you for submitting your article "Dynamics of gaze control during prey capture in freely moving mice" for consideration by *eLife*. Your article has been reviewed by three peer reviewers, and the evaluation has been overseen by a Reviewing Editor and Laura Colgin as the Senior Editor. The following individuals involved in review of your submission have agreed to reveal their identity: Tadashi Isa (Reviewer #1); Anthony Leonardo (Reviewer #2); Anne K. Churchland (Reviewer #3).

The reviewers have discussed the reviews with one another and the Reviewing Editor has drafted this decision to help you prepare a revised submission.

Summary:

This paper presents a creative task and utilizes an elegant, wearable two-camera measurement device along with a 3-d accelerometer and gyroscope to investigate eye and head movements in free-moving mice during general exploration and natural prey capture. The authors show that the majority of eye movements are compensatory for head movements which stabilizes the visual scene. At the same time, they also recorded non-compensatory saccades. However, head movements are targeting the prey more precisely than the eyes. In a nutshell, head movements are the primary gatherer of directed visual information, and eye movements are used in a compensatory fashion. This is broadly consistent with studies on other vertebrates that lack a fovea and overall what one might expect for a fovea-less organism (no need to orient eyes except for keeping visual target of interest broadly within them; see for example Borghuis' study on amphibians which shows much the same). The authors also measured vergence. They found that during approach of the prey, the animal's head tends to be at a neutral pitch, and neutral pitches are associated with neither convergent nor divergent eye movements. Taken together, the authors argue that these observations constrain the mouse's oculomotor strategy during prey capture, and that this strategy can be summarized as "saccade and fixate."

The reviewers and I agree that the experimental study and data analysis are technically rigorous and the paper is rich with detailed statistical information. Whereas there is no fundamentally new result here about gaze control on a general computational level, this work will be of great use to the mouse vision and behavior community. There is also considerable current interest in measuring neural activity during free-moving behavior making this an important paper in terms of its rigorous characterization of the accompanying head and eye movements. Further, the authors selected a really nice behavior, cricket hunting, that highlights the dual role of mice as not only prey but predator as well. This dual role means that the lateral positioning of the eyes on the mouse's head is somewhat mysterious, raising questions about what its oculomotor strategy might be during prey capture.

All reviewers emphasize the importance of discussing in more detail the limitations of a measurement device that samples eye movements at 30 Hz. I endorse these comments and would like to see them addressed in detail in a revision.

1) The authors recorded the eye movements with the time resolution of 30 Hz. This is obviously slow to record the rapid eye movements in general. In case of primate and human studies, the standard is 1000Hz. In case of rodents, some older studies recorded at 240 Hz and the results showed that under the head-fixed condition, the peak velocity of saccades in mice was several hundred degree/s (Sakatani and Isa, 2004, 2007). Reviewer #1 is concerned that the authors missed the very fast component of saccades from their analysis and concluded the dominance of compensatory eye movements, which are slow.

2) The authors' system for measuring eye position is well-suited for freely moving animals. Reviewer #3 would have liked to see a bit more data validating that system. For instance, the authors might compute the gain of the VOR and see how its mean and variance measure up to other studies. This would be somewhat tricky since most other work is in head-fixed animals, but it would at least give the reader a sense as to how the measures here compare to others in the field.

3) Along the same lines, more plots of eye velocity and not just position (e.g., in Figure 1) would be helpful. I recognize that with a 30Hz camera sampling rate, the velocity traces will be noisy; nonetheless, it would help give readers a sense of how the measurements here square up with others. Because this system will likely be used by other groups, due to its many advantages, having a sense of any pitfalls will be helpful.

Other methodological questions:

4) The authors need more explanation about the GMM clustering and how they divided compensatory eye movements and saccades in the Materials and methods section. In Figure 4A, it seems some red dots and blue dots are mixed with each other.

Terminology / Discussion:

5) Please provide clarification regarding the pursuit terminology. Whereas the term "pursuit" might be standard in interception and behavior studies of predator/prey interactions, with many decades of use, there is some ambiguity with smooth pursuit eye movements – slow, continuous rotations of the eyes to keep small moving objects close to the fovea. Reviewer #2 suggests additional editing and clear definitions at the beginning of the manuscript (e.g. prey pursuit vs smooth pursuit).

6) Discussion: Reviewer #3 was surprised that the authors described mice as, "not a highly visual species." Their own work highlights that mice use vision for prey capture. Further, as the authors, know, mice have multiple (~7) independent maps of visual space in the brain. Why would the brain go to the trouble of generating all these maps if mice didn't make much use their visual systems? Finally, describing an animal as "highly visual" or "not highly visual" ignores the fact that all animals use multiple sensory systems to navigate the world, and tradeoff which sensory modality is weighted most heavily in a dynamic way (e.g., relying on vision during daytime hunting/foraging and somatosensation at night).

---

## [Author Response]

Revisions for this paper:All reviewers emphasize the importance of discussing in more detail the limitations of a measurement device that samples eye movements at 30 Hz. I endorse these comments and would like to see them addressed in detail in a revision.1) The authors recorded the eye movements with the time resolution of 30 Hz. This is obviously slow to record the rapid eye movements in general. In case of primate and human studies, the standard is 1000Hz. In case of rodents, some older studies recorded at 240 Hz and the results showed that under the head-fixed condition, the peak velocity of saccades in mice was several hundred degree/s (Sakatani and Isa, 2004, 2007). Reviewer #1 is concerned that the authors missed the very fast component of saccades from their analysis and concluded the dominance of compensatory eye movements, which are slow.

We greatly appreciate this concern, which spurred us to re-analyze our data at higher framerate, an option we had not pursued previously. In order to achieve this, we have taken advantage of the fact that our 30Hz video was in fact in interlaced format, meaning that there were actually two sequential images in each video frame, interleaved together in alternate horizontal rows. By de-interlacing these two images, we are able to restore the full 60Hz video rate, albeit at a 2x loss of spatial resolution along the y-axis. This loss of spatial resolution does not impede our ability to resolve pupil position (estimated at <=0.5deg as discussed below), and therefore *we* now present all our results at 60Hz temporal resolution, which has been standard in several recent studies of eye movements in freely moving rat and mouse. While this provides improved segregation of non-compensatory movements (Figure 4B) and clearer demonstration of the saccade-and-fixate pattern (Figure 4C), our overall findings are unchanged. We greatly appreciate the motivation to improve our data quality in this manner. We have also added a discussion of the temporal resolution and comparison to studies using various methods.

2) The authors' system for measuring eye position is well-suited for freely moving animals. Reviewer #3 would have liked to see a bit more data validating that system. For instance, the authors might compute the gain of the VOR and see how its mean and variance measure up to other studies. This would be somewhat tricky since most other work is in head-fixed animals, but it would at least give the reader a sense as to how the measures here compare to others in the field.

We appreciate the need to provide benchmarks for validating our system, and we now include assessment of the accuracy of the resulting measures as discussed below in specific points raised by the reviewers. In particular, we provide estimates of error in measurement of eye and head positions, as well as camera jitter, in the Materials and methods. These factors are all <1deg, similar to other recent studies in freely moving rodent vision. This also supports the robustness of results presented, though it represents a limitation in the smallest movements we might detect, such as micro-saccades. Furthermore, these benchmarks provide a direct measurement of the performance of the system and do not require comparison to head-fixed recordings such as for VOR gain. Finally, we note that the basic parameters of eye movements, amplitude and velocity, agree with recent comparable studies in freely moving animals.

3) Along the same lines, more plots of eye velocity and not just position (e.g., in Figure 1) would be helpful. I recognize that with a 30Hz camera sampling rate, the velocity traces will be noisy; nonetheless, it would help give readers a sense of how the measurements here square up with others. Because this system will likely be used by other groups, due to its many advantages, having a sense of any pitfalls will be helpful.

We have now included eye velocity traces in Figures 1, 4, and 5. These are not as noisy as one might have expected, particularly with 60Hz resolution, and certainly do help demonstrate the pattern of eye movements more clearly.

Other methodological questions:4) The authors need more explanation about the GMM clustering and how they divided compensatory eye movements and saccades in the Materials and methods section. In Figure 4A, it seems some red dots and blue dots are mixed with each other.

Based on the concern about the clarity of the GMM clustering, as well as the improved temporal resolution resulting from 60Hz video data, we have simplified our segregation of compensatory and non-compensatory eye movements. We now use a threshold on gaze velocity, similar to that frequently applied to eye velocity in primate studies, which is supported by the distribution of eye/head movements in Figures 4A,B.

Terminology / Discussion:5) Please provide clarification regarding the pursuit terminology. Whereas the term "pursuit" might be standard in interception and behavior studies of predator/prey interactions, with many decades of use, there is some ambiguity with smooth pursuit eye movements – slow, continuous rotations of the eyes to keep small moving objects close to the fovea. Reviewer #2 suggests additional editing and clear definitions at the beginning of the manuscript (e.g. prey pursuit vs smooth pursuit).

We recognize that we were unclear in using the terms pursuit and approach interchangeably, particularly given the potential confusion with “smooth pursuit”. We now use the term approach throughout.

6) Discussion: Reviewer #3 was surprised that the authors described mice as, "not a highly visual species." Their own work highlights that mice use vision for prey capture. Further, as the authors, know, mice have multiple (~7) independent maps of visual space in the brain. Why would the brain go to the trouble of generating all these maps if mice didn't make much use their visual systems? Finally, describing an animal as "highly visual" or "not highly visual" ignores the fact that all animals use multiple sensory systems to navigate the world, and tradeoff which sensory modality is weighted most heavily in a dynamic way (e.g., relying on vision during daytime hunting/foraging and somatosensation at night).

We agree completely, as the reviewer infers from our previous work. We simply meant to refer to the lower visual acuity relative to some other species that are commonly studied in visual neuroscience. We have now removed this phrasing.